# Exploring and prioritising strategies for improving uptake of postnatal care services in Thyolo, Malawi: A qualitative study

**Alinane Linda Nyondo-Mipando** [1]*, **Marumbo Chirwa**[1], **Sangwani Salimu**[1], **Andrew Kumitawa**[2], **Jacqueline Rose Chinkonde**[3], **Tiyese Jean Chimuna**[3], **Martin Dohlsten**[4], **Bongani Chikwapulo** [5], **Mesfin Senbete**[3], **Fatima Gohar**[3], **Tedbabe D. Hailegebriel**[6], **Debra Jackson**[7,8]

1 Department of Health Systems and Policy, School of Global and Public Health, Kamuzu University of Health Sciences, Blantyre, Malawi, 2 Department of Epidemiology and Statistics, School of Global and Public Health, Kamuzu University of Health Sciences, Blantyre, Malawi, 3 UNICEF Malawi Country Office, Lilongwe, Malawi, 4 Department of Maternal, Newborn, Child and Adolescent Health and Ageing, World Health Organization, Geneva, Switzerland, 5 Quality Management Directorate, Ministry of Health, Lilongwe, Malawi, 6 Health Section, Programme Division, UNICEF Headquarters, New York, NY, United States of America, 7 London School of Hygiene and Tropical Medicine, London, United Kingdom, 8 University of the Western Cape, Cape Town, South Africa

* lmipando@kuhes.ac.mw

**Data Availability Statement:** The datasets used and/or analysed during the current study are all included in the manuscript as part of the results.

## Abstract

Although postnatal care services form a critical component of the cascade of care in maternal, newborn, and child health the uptake of these services has remained low worldwide. This study explored and prioritised the strategies for optimising the uptake of postnatal care (PNC) services in Malawi. A qualitative descriptive study followed by nominal group techniques was conducted at three health facilities in Malawi from July to December 2020 and in October 2021. We conducted focus group discussions among postnatal mothers, fathers, healthcare workers, elderly women, and grandmothers. We conducted in-depth interviews with midwives and key health managers. Nominal group techniques were used to prioritise the main strategies for the provision of PNC. The demand strategies include appointment date reminders, provision of free health passport books, community awareness campaigns, and involvement of men in the services. The supply strategies included training health providers, improving clinic operations: task-shifting and hours of operation, having infrastructure for the services, and linkage to other services. Having services delivered near end-user residences was a crosscutting strategy. Refresher training and improvement in the clinic operations especially on hours of operation, appointment date reminders, and linkage to care were the prioritised strategies. There is a need to use acceptable and contextualised strategies to optimise the uptake and delivery of postnatal care services. Educating the healthcare workers and the community on postnatal services is key to increasing the demand and supply of the services.

**Funding:** The study was supported financially by UNICEF New York, Grant number 43292967. ALNM, MC, SS, and AK received support through the grant. JRC, TJC, MS, FG and DJ were supported by UNICEF, MD was supported by WHO and BC was supported by the Malawi Ministry of Health. The funders had no role in study design, data collection and analysis, decision to publish, or preparation of the manuscript. The content is solely the responsibility of the authors and does not necessarily represent the official views of UNICEF.

# Introduction

Postnatal care (PNC) services form a critical component of the cascade of care in maternal, newborn, and child health [1]. The World Health Organisation (WHO) advocates for the provision of PNC services to mitigate neonatal and maternal deaths [1]. Globally, the rates of maternal and neonatal deaths postnatally remain unacceptably high with 30% of maternal deaths occurring during the postpartum period while 17 in every 1000 liveborn babies die in the first month after birth [1] underscoring the relevance of PNC services. In Malawi, 65% of maternal deaths occur in the postpartum period [2] while most infants die in the neonatal period with a neonatal mortality rate of 26 deaths per 1000 livebirths whereas the postneonatal mortality rate is at 14 deaths per 1000 livebirths [3]. According to the recent Maternal Death, maternal sepsis (25%), eclampsia (20%), hemorrhage (13%) are the main causes of maternal deaths in Malawi. Inadequate monitoring of a woman in labor, delays in acting and exposing a woman to prolonged abnormal observations, and inadequate emergency skills are additional factors related to health workers [2]. Additional factors include delays in receiving health care, communication and transportation difficulties, and supply shortages [2].

Postnatal care services are to be provided within 24 hours of birth, 48–72 hours, 7–14 days, and six weeks after birth and include various recommended services with context-specific adaptations [1]. Within 24 hours of giving birth, moms are assessed for the amount and flow of vaginal bleeding, uterine contraction, blood pressure readings, fundal height, temperature, and pulse. Mothers are questioned about micturition, urine incontinence, bowel function, the recovery of a perineal tear, headache, exhaustion, backache, perineal pain and hygiene, breast and uterine soreness, and lochia at the subsequent PNC exams [1]. Every time a woman receives postpartum care, her emotional wellness, and the support she receives are evaluated, and she is urged to report any changes to her mood, emotional well-being, or behavior. A woman is given information on family planning techniques and symptoms of postpartum hemorrhage [1]. WHO further recommends that HIV catchup testing and screening for Tuberculosis be done in contexts where the risk is high. A newborn is evaluated at each postpartum visit if they have stopped eating well, have a history of convulsions, are breathing more than 60 breaths per minute, have severe chest contractions, are not moving on their own, have a temperature higher than 37.5 degrees Celsius, a low body temperature lower than 35.5 degrees Celsius, have yellow palms and soles at any age [1]. WHO further recommends for assessment of hearing impairment, eye disorders, and hyperbilirubinaemia [1].

Malawi adopted the same assessments as above and provides postnatal care in primary, secondary, and tertiary care facilities, with most services being provided in hospitals [4]. Furthermore, PNC should be given three times: the first one, immediately after delivery in a medical facility, the second time, one week later, and the third time, six weeks later, when puerperium ends, and this is irrespective of the place of delivery [4].

A major determinant of PNC service provision is the length of stay in the facility post-delivery which varies across countries. A review from low- and middle-income countries showed high variation in the length of stay post-delivery, ranging from half a day to 9 days with further variations influenced by the nature of birth [5]. There is a dearth of information on studies that specifically addressed contextually driven strategies that could be implemented in Malawi.

In Malawi, despite a high proportion of women visiting health facilities for antenatal care (ANC) (97%) and delivery care (97%), indicators of access to postnatal care (PNC) lag with the rate of PNC care for mothers and newborns within the first 48 hours of delivery at 84% and 88% respectively [3]. The rate of Postnatal care falls miserably at one and six weeks of visits to 30.2% at one-week postdelivery and 6.1% at six weeks for the mother while for the baby it is at 78% at one week of age and 14% at six weeks of age [6]. Evidence shows that barriers to PNC

utilization included rural residency, young age group, low education level, unemployment, non-exposure to media, and non-attendance to ANC, coupled with limited access to health services [7]. A review showed that the provision of PNC services is influenced by the training and supervision of healthcare workers, workload of a facility, remuneration, living conditions of healthcare workers, availability of and access to well-equipped, well-organized healthcare facilities with amenities like water, electricity, and transport, and communication between health care workers (HCWs) and mothers [8].

The strategies can be on the demand side as per the end-users' preference or can be on the supply side which includes those that stem from the health facility. The demand strategies in place for optimising the uptake of PNC services include community awareness of the importance of PNC [9] and offering PNC services in the communities where women reside [10, 11]. The supply-side strategies include task shifting of services to community-based health workers [11, 12], and training of HCWs on the required assessments to be delivered [10]. Given the relevance of PNC in curbing maternal and neonatal mortalities coupled with the lower rates of coverage of PNC visits [1], and recent calls to find strategies to improve the uptake of PNC services [13], there is a need to identify contextual strategies that optimise the uptake of PNC care in Malawi. This study, therefore, explored and prioritised the strategies for optimising the uptake of PNC services for the mother and baby in Thyolo, Malawi. This information will inform the local teams in Malawi and beyond on strategies that they can adopt to improve the services.

## Methodology

### Study design

A qualitative descriptive study was conducted from July to December 2020 in Thyolo, Malawi. After analysing the qualitative findings, nominal group techniques (NGTs) [14] were conducted in October 2021 to prioritise strategies for optimising the uptake of PNC services. For the participants to provide specific details derived from their experiences, we included a purposively selected sample of women who were postnatal mothers and health care professionals who were involved in providing postnatal care. We conducted focus group discussions (FGDs) among postnatal mothers, fathers whose partners were postnatal mothers, HCWs including Health Surveillance Assistants (HSAs), elderly women, and grandmothers. We conducted in-depth interviews (IDIs) with midwives and key HCWs. Participants for NGTs included postpartum women, fathers, midwives, clinicians, health surveillance assistants who are providers of community-based maternity services, and managers who oversee the provision of PNC services in Thyolo.

### Study setting

The study was conducted in three facilities in Thyolo District: a health centre, a rural hospital, and the district hospital. Thyolo is in the southern region of Malawi and the 2018 census showed that it had a population of 458,976 people, 177, 298 were women of childbearing age [15]. The facilities included a health center 20 kilometers from the district hospital (Facility 1), a rural hospital 53 kilometers from the district hospital (Facility 2), and the district hospital (Facility 3), which also serves as the district's referral hospital for all other health centers and hospitals. The selection of the facilities was varied according to geographic area and functionality so that we reach richer descriptions of the strategies in the district and at varied levels of health service provision. Facility 1 is located at a vibrant trading centre and is the commercial hub of the district while Facility 2 is in a rural area with mountainous terrain. Thyolo was selected as a research site because it had persistently registered poor maternal and newborn

indicators in the four years preceding the study. All facilities provide maternity services with only the district hospital conducting caesarian section operations. Unlike the other facilities, which primarily employ nurse-midwife technicians and medical assistants, Facility 3 has a diverse range of healthcare professionals who provide maternity services. These professionals include nursing officers, clinical officers, and medical officers, who have training at the bachelor's and master's levels.

## Sampling and sample size

We drew a purposive sample and maximized variations of our respondents to achieve a broader scope of responses [16]. Our goal was to have a sample that would enable us to obtain the rich data required for every technique of data collection. We performed seven in-depth interviews in total, with the midwives who provided care in the facilities and the nursing and midwifery in charge of the maternity wards in all three facilities. Saturation of ideas [17] as well as information power by ensuring the selection of participants with a depth of information in PNC services, specifying the roles they play, and the data collection approach of engaging them in the dialogue [18] guided the data collection approach to achieving an adequate sample size. In the focus group discussions, we settled for a focus group discussion per site per group of people and held a total of 12 focus group discussions, translating into four FGDs per site. We believed that with 12 FGDs, we would reach 90% of the themes as suggested by Guest et al. [19]. The sampling techniques and sample size are illustrated in Table 1. The total number of participants for NGTs was 33 and included six postpartum women and five fathers with babies under the age of 6 months, 11 nurses-midwives, six HSAs, three health Managers, and two clinicians. HSAs are a type of community-based health worker whose primary responsibilities include the provision of preventive health services like immunization, health education, and the promotion of hygiene and sanitation. They also provide family planning, child health, nutrition, and community-based maternal and neonatal care services [20, 21]. We included healthcare workers who were willing to participate in the study, work in maternity, postnatal, or as HSAs, and were 18 years of age and older.

**Table 1. Sample size and sampling technique for qualitative component.**

| Participant type | Method of data collection | Number | Sampling Technique | Rationale |
|---|---|---|---|---|
| Maternity Staff including a Unit Matron (A Nurse/Midwife The manager overseeing the maternity services at a hospital), Postnatal Ward In charges (Nurse-Midwives with the responsibility of leading and managing the unit)-, Maternity Ward in Charge (Nurse-Midwives with the responsibility of leading and managing the unit), Nurse-Midwives | In-depth Interview | 7 | Purposive | These were key to the provision of PNC services at various levels. |
| Elderly women and Grandmothers | Focus Group Discussions | 24 | Purposive | They advise women on maternity aspects |
| Men | Focus Group Discussions | 24 | Purposive | They are decision-makers in families and provide for resources to access care |
| Health Surveillance Assistants | Focus Group Discussions | 24 | Purposive | They provide Postnatal Care services in static and outreach centres. They are liaison points with the community |
| Postnatal Mothers | Focus Group Discussions | 24 (Varied with age) | Purposive (deliberately feature mothers of varying attributes of interest | These are the beneficiaries of the services. |

We excluded students from various institutions and those who had worked for less than a month in postnatal care settings. We included men in FGDs who were available and willing to participate, and those with a baby who was less than a year old, as determined by asking for the child's health passport book. We included elderly women who were responsible for counseling women on maternity aspects, recognised as resource persons for maternity aspects in a community, and grandmothers of grandchildren under 12 months of age. We included postnatal women who were willing to participate in the study, postnatal women with or without a living baby. If with a living baby, the age of the baby was varied: immediately after birth, 24 hours, 48 hours, 7 days, and 42 days after birth; we excluded postnatal women who were physically and mentally unstable mothers.

## Recruitment of study participants

Fathers and community members were recruited with the assistance of the health care workers in the areas of their residences. The study team shared with the healthcare workers the inclusion criteria that were used to identify the potential participants. Upon identification of eligible men and elderly women for FGDs, they were booked to attend a discussion on a specific day and were asked to report to the facility for the FGDs. Mothers were recruited at the clinics as they presented themselves and were booked for an FGD later. If the numbers were not enough health care workers identified other women to attend the FGDs. Health workers were recruited by the research assistants while following the eligibility criteria. Mothers, fathers, and healthcare workers for NGTs were recruited with assistance from the facility in charge after presenting the criteria to them. The research Assistants verified the eligibility of each participant before recruiting them for the study.

## Data collection

**Qualitative data.** All data were collected face-to-face, and audio-recorded, and field notes were captured as well. FGDs were deliberated in Chichewa (local language) while Interviews were in English using an interview guide following pretested topic guides (S1, S2 Texts). Before data collection, the topic and interview guides were tested on a comparable population that was found at Facility 1 to ascertain their ability to gather the information that was needed.

The findings from the pilot phase were not included in the overall results however they highlighted areas that needed revisions to allow for more probing. All participants had an assigned code that was used for identification during the FGDs which were held in a private setting that was determined by the participants with only researchers and participants present. The FGDs were separated into groups based on age and sex. There were three FGDs for men, three for elderly women, and three for postpartum women. Three of the four research assistants in the study were females. All research assistants were experienced and trained in the collection of qualitative data. Research assistants had no prior knowledge of the participants and they introduced themselves as research assistants in a project. None of the participants refused participation nor were there any repeat interviews. Interviews and discussions averaged 35 and 60 minutes respectively. Data collection stopped when there was a saturation of ideas.

**Quality of qualitative data.** At the end of each interview and discussion, all the key points from the session were summarized for the participants to verify the findings as a measure of member checking [22]. We have included the participants' quotes in our results to achieve the dependability of our findings [22]. The credibility of our findings was increased by gathering information from a variety of sources, including healthcare professionals, postpartum women, elderly women, and male partners, and using a variety of data collection techniques, including FGDs, observations, and in-depth interviews [22]. We maximised the transferability and

confirmability of our findings through a detailed description of the setting where we conducted the study and the explanation of the methods that we followed thereby creating an audit trail for others to follow [22]. The research team considered their knowledge of postnatal care services and experiences to appropriately handle reflexivity and prevent personal experiences from distorting data collection and processing. This was accomplished through iterative team meetings held during the data collecting and analysis period, which were aided by each researcher documenting their knowledge, beliefs, and experiences related to postnatal care.

**Nominal group technique.** Following an initial analysis of the qualitative data, we held two consensus meetings that utilized NGTs among the various participants. The consensus meetings were split into two and included one with health centre-level participants and the other with hospital-level participants. The NGT participants in the health centre group included six postnatal mothers and five fathers with babies under 6 months of age, three HSAs, and six nurses and midwives. The hospital-level facility includes five nurse-midwives, three health managers, two clinicians, and three HSAs.

We planned on having one session, but it proved difficult to assemble all the participants in one place due to work commitments. In both meetings, the sessions started with a presentation of the findings on the identified strategies from the interviews and focus group discussions. The steps taken for the NGTs are outlined in Fig 1. All participants first heard the strategies, together with a summary that was provided on paper, in a local language. To prevent power dynamics that might have arisen due to the positions of the healthcare workers, the initial groups were split such that the women and men each had their own group. Those who couldn't write expressed their preferences verbally, and a research assistant MC would write them down on the paper for them. To facilitate a productive conversation that did not disadvantage individuals who could not read or write, the discussions were also held in a local language and were only translated into English at the point of data transcription.

Stage 1: Results on the strategies were shared to the attendants and they were provided with a summary of the strategies on a piece of paper in both English and Chichewa languages

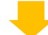

Stage 2- Silent Generation: Participants were allowed to reflect and make their own prioritisations of the strategies

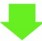

Stage 3- Group Generation and Round Robin (an act of going around in the group for each participant to share his/her ideas as they are): Participants were assembled in their groups as per Table 2 and were allowed to discuss in their groups on the prioritised strategies they compiled at an individual level and develop a groups' priority list

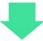

Stage 4- Round Robin and clarification: Each group presented their priority list to the bigger group as was in Stage 1. Clarity was sought on the priority list for groups to justify their lists

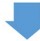

Stage 5: Consensus agreement on the best three strategies by the whole group

**Fig 1. Steps during the nominal group techniques.**

**Data management and analysis.**   All data were kept in locked cabinets at Kamuzu University of Health Sciences (KUHeS) and in password-protected computers with access limited to lead researchers. Data were managed using NVivo 12 software (QSR International, Melbourne, Australia) and audio recordings were transcribed verbatim and translated into English as applicable. A coding guide was inductively and deductively developed by two researchers after reading two transcripts each followed by a discussion to harmonize codes. A coder applied the coding guide to all transcripts and the coding process was reviewed by ALNM by coding a selected number of transcripts to ascertain the application of the coding guide. MC, SS, and ALNM held multiple meetings to review the coded data and discussed the new codes as they came up from the data. Data were analysed using a thematic approach where similar codes were grouped under overarching themes [23]. The themes were reviewed for adequacy and correct representation of data and were checked against the audio to ensure that there was no misrepresentation of findings. The themes with limited data were combined with other similar or related themes while ensuring that the themes remained stand-alone without overlaps. The themes were iteratively discussed among the researchers to achieve clarity in data presentation.

The stages in the nominal group technique are illustrated in Fig 1. At each of the stages, there was a development of a list at both the individual level and group level prioritization. The process of prioritising the strategies at the health centre and hospital as well as the consensus meetings were facilitated by the researchers. The prioritised lists from the various groups were presented during the consensus meetings and were discussed by all members to reach an agreement. The team members weighed the feasibility of the strategies and agreed on three priorities based on the identification of short and long-term strategies. The discussion also revolved around resources needed for the facilities to implement the strategies and included those that did not require insurmountable resources and were also deemed to be within their control as a short-term priority, while strategies that required the involvement of stakeholders outside of their areas were deemed as long-term priority strategies.

## Ethics approval

The study was reviewed and approved by the KUHeS' ethical committee (CoMREC No P.03/20/2977). All participants gave written informed consent before data collection. Participants who could not read or write thumb printed the form in the presence of an impartial witness. Participants in the NGTs provided verbal consent before starting the meetings. This was obtained following a description of the meeting and the procedures that were to be followed in the meeting. Participants were assigned codes that were used throughout the data collection and management process to maintain confidentiality. Only consent forms had information that identified the participants and were immediately stored in a secure place at KUHeS as per CoMREC's regulations.

## Results

### Characteristics of Health Surveillance Assistants

In total, we had 24 HSAs that attended FGDs. There were 3 FGDs and each group comprised 8 HSAs. The age of the HSAs ranged from 23 to 49 years old and of the 24 HSAs 12 were females. All of them had attained a secondary school education and had been in service for 6 to 27 years.

**Characteristics of nurse-midwives.**   Of the seven nurses-midwives interviewed four were Registered Nurses Midwives, five were either facility or departmental in-charges, and six were

female. Their age ranged from 24 to 36 years. The nurses-midwives served for a period from six months to nine years.

**Demographic characteristics of postnatal women, men, and elderly women.** The demographic characteristics of men, postnatal women, and elderly women are presented in S3 Text. Most of the men ran small businesses, most of the elderly women were farmers, and most of the postnatal women were housewives.

## Strategies for improving uptake of PNC Service

The strategies that were suggested are categorized from the supply, and demand side of services, and a crosscutting theme of Proximity of Clinics to clients. The demand strategies are the ones that end-users want to have in place while the supply strategies are those that a health system needs to put in place (Fig 2).

**Demand-side strategies.** The demand strategies include appointment date reminders, provision of free health passport books, community awareness campaigns, and involvement of men in the services.

**Appointment date reminders.** Healthcare workers stated that they need to always let the woman know the next date they need to report for PNC services. It was reiterated that at times a postnatal woman may not know their next date of appointment and hence may not show up for PNC services.

*"I think we [health care workers] should remember to tell them [PNC women] the date for the next visit. This will encourage them. Let's say they visit on the 27th of this month then we tell them to come on the 27 of the next month. This will encourage them to come."* Facility 3 HSA FGD

**Demand Side Strategies**: appointment date reminders, provision of free health passport books, community awareness campaigns, and involvement of men in the services

**Supply Side Strategies:** training health providers having infrastructure for the services, linkage to other services and improving clinic operations (task-shifting and hours of operation)

**Crosscutting Strategy:** Proximity of clinics to areas of residence

**Fig 2. Summary of themes and subthemes of the proposed strategies.**

**Provision of free health passport books.** The provision of free health passport books is another strategy that can be used to improve attendance to PNC services. This strategy has been applied before and participants stated that it encouraged women to use PNC services.

"*Some things which can make them come back is receipt of a free health passport book for the mother and baby. In the past, the babies were receiving a card, health passport for free and the mother was also receiving one for free which has been stopped, but that encouraged them to attend postnatal services early.*" Facility 2 HSA FGD

**Community awareness.** Within the community, participants suggested conducting awareness campaigns as the main strategy for improving the uptake of PNC services. These sessions could be in the form of education services and posters that could be displayed in the communities. These sessions should be linked to the bylaws that a chief can apply to promote adherence to the sessions. Participants stated that chiefs are key to the delivery of such sessions and will uphold the messages and ensure that their people are compliant. Religious leaders should be involved because they also influence the uptake of PNC services.

"*The major strategy is teaching people in the villages. Maybe starting with the people who have positions, the chiefs, those with different positions in the villages. Then they should teach the people in their villages and teach the health workers in the areas so that they can be encouraging them.*" Facility 1 Healthworker IDI

"*They [health care workers] must promote messages about attending postnatal care. These messages will deal with some beliefs that are against postnatal care. They must be encouraged through different messages about postnatal care like they do when they discourage women from delivering in their homes. The penalty enacted by chiefs of paying a goat when one delivers at home should also be extended to postnatal care.*" Facility 3 Men FGD

**Drama groups.** Elderly women emphasized the need for drama groups that will offer the necessary message on PNC services theatrically as a measure of encouraging women to comply with PNC services.

"*They use the support groups who dramatise events, and stage some role-plays to show people the importance of safe motherhood which will teach and encourage people to attend postnatal care services. The government should be sending these groups so that while they are here to entertain us people will also learn important things.*" Facility 1 Elderly Women FGD

**Health mentors.** Elderly women reflected on being exemplary to the younger women and encouraging them to attend PNC services. Other health mentors could be from outside the community who can visit the community and encourage mothers on the importance of PNC Services.

"*The young mothers themselves also need encouragement from us [elderly women] for without the encouragement they may not come; the health mentors need to be coming here frequently to encourage the mothers to attend PNC.*" Facility 1 Elderly Women FGD

**Male involvement.** Male involvement facilitates the utilization of PNC because women that attend PNC services with male partners are prioritised at the facilities as per common practice in Malawian facilities. Male involvement in PNC services is an opportunity for men to learn more about what PNC services are about and for them to better support their partners. Men acknowledged their roles in sourcing transport for the women to attend PNC services.

*"The messages about health should also reach men so that men should encourage women if they are lazy to attend postnatal service because men have the decision-making authority and can cause a woman to attend or miss postnatal care services."* Facility 2 PNC Women FGD

**Supply-side strategies.** The supply strategies included training health providers, improving clinic operations: task-shifting and hours of operation, having infrastructure for the services, and linkage to other services.

**Training of health providers.** Healthcare workers suggested that there is a need for HCWs to be refreshed on the appropriate assessments and time points for postnatal care. It was suggested that training should reinforce skills in assessing a postnatal woman and her baby because HCWs rarely receive any refresher training in PNC while in service. Participants deemed that the refresher course would assist HCWs who have not worked in a postnatal unit for a long time.

*"Checking on the woman procedurally are things that we learn at school so if you notice some left school way back. Maybe they were working in the ward where they could just attend to people who are suffering from malaria and these other basic diseases so if they are coming to this ward, they may have forgotten some of the things."* Facility 3 Health Worker IDI

*"So, what we need is most people to be trained so that when they are trained, they can implement. . .of course, we are trained at school, but we need to be refreshed because there are a lot of things that we are implementing."* District Safe Motherhood Coordinator IDI

**Improving clinic operations.** Healthcare workers and the other participants asserted that another strategy to improve the delivery of PNC services would be a change in some of the clinic operations. These changes included task-shifting PNC assessments to other cadres of health workers, extending the opening hours, and linkage to other services.

**Task shifting and sharing of responsibilities.** Participants suggested having a cadre that could take on some of the burdens that are handled by nurses and HSAs in delivering PNC services. This will lighten the workload among nurses and this cadre, which comprises lay people, could be trained in what they ought to do and focus on those areas that are not very technical such as delivering health education messages and assisting women to understand the flow of the clinic. Participants suggested that HSAs could be trained in the provision of less complicated aspects of PNC as part of their community-based maternal and neonatal care.

*"Another strategy, for everyone working here at the hospital whether a guard or cleaner, should be taught so that they can be delivering the message, aiding the women [to understand the flow at the clinic]. I feel this can also help because when everyone will be taking part . . . They can also be asking after receipt of specific services if one has accessed PNC services."* Facility 1 Health worker IDI

*"As HSAs, we were taught so well about a home visit, for antenatal and postnatal, so it is the duty of every HSA in his or her village because everyone has his or her village, to visit every woman . . ."* Facility 1 HSA FGD

**Hours of operation.**   Mothers stated that the facilities should open up for PNC services early enough so that women are not kept for a long time at the facility. Participants reported that this visit will enable those who are involved in small-scale businesses to attend to their trade after the PNC visits.

*"They also should be opening as early as possible so that we should also be encouraged to be punctual knowing if we are late, we will find them done."* Facility 1 PNC Women FGD

**Infrastructure.**   Healthcare workers stated that having adequate rooms within a facility where PNC services could be offered is another strategy for optimising PNC uptake. If there are adequate rooms, mothers and their babies will not wait for a long time, and it would also enable HCWs to examine them properly.

*"If we had a room as we do for services like family planning. . . we do not have a room for examining women who have delivered or those that have reported with their babies for the one-week postnatal visit or six weeks. We just use a temporary space in the same labour ward or the postnatal room."* Facility 1 Health Worker IDI

**Linkage to other services.**   For women to know where and when to obtain the services they would need in the postnatal period, healthcare workers advised increasing the links to the services that are already available. They emphasized the significance of explaining to a lady who has arrived for immunization services why it is important for her to attend postnatal care. They further stressed strengthening the link between the health facility and community systems so that an HSA should be able to get a report of those who have delivered that reside in his or her catchment area for proper follow-up in the area. They went on to say that follow-up visits in the postnatal period will be simple to implement if an HSA has been told of a new birth or about a postnatal mother and if the woman has been informed of the HSA in her area.

*". . .the nurse who helped the mother deliver should notify the HSA of the area where the mother came from so that she can be followed up."* Facility 2 HSA FGD

*"This issue of Community-Based Maternal and Newborn Health (CBMNH) which is for HSAs to help women. . . HSAs will document what they have done . . . HSAs will be able to show that this woman was assessed, her issues highlighted, and the measures they took to address the issues. . ."* District Safe motherhood IDI.

Mothers also asserted that HSAs should be linked up with the women in their catchment areas to promote PNC uptake.

*"HSAs in our community must be in touch with women which ultimately encourages them to come and access postnatal care. But when there are no HSAs, most women engage themselves in business and farming forgetting about postnatal care. So, when HSAs are encouraging*

*women, it becomes easy for us [Postnatal Mothers] to come here [Health Facility].*" Facility 1 PNC Women FGD

The linkage to care between a health facility and the community could be better served if there were registers that are used for follow-up purposes whenever one conducts a home visit.

"*Some health workers were taught that if a woman has delivered, they should be visiting her at her home. This strategy is good, but it needs to have evidence that health workers are working. Yes, visiting the mother, maybe having a Register that captures details of the postnatal women in an area. Then everyone should be able to follow up on those women.*" Facility 1 Health Worker IDI

**Crosscutting strategy.**   A cross-cutting strategy that was suggested was having clinics closer to users' areas of residence. This was raised on the demand as well as supply side.

**Proximity of clinics to clients.**   Participants advocated for having clinics closer to where mothers reside to ease the burden of distance and associated transportation costs to the facility.

"*I feel like the best way, health officials need to run clinics in the villages so that for those who cannot manage to visit a hospital they can be reviewed within the village. . . there is a need to build clinics close by so that we can have access to medical services.*" Facility 1 PNC Mothers FGD

"*We do mobile clinics in some facilities because the facilities are very far, and we need to make sure that the women are not covering a long distance for them to get to a facility.*" District *Safe* Motherhood IDI

**Prioritisation of strategies.**   Following NGTs, the strategies were prioritized at the individual and then the group levels which were later consolidated to achieve a consensus prioritised list.

**Individual-level strategies at the health centre and hospital level.**   Five of the 19 individuals in the health centre group prioritised the training of HCWs while another five selected the presence of dedicated infrastructure for the provision of PNC services as the key strategy. Nine out of 14 of the hospital-based participants selected training of HCWs on what they ought to do during PNC visits as a key strategy, followed by five that listed a dedicated infrastructure and six listed having clinics near where the people were, as the third common strategy.

**Group-level prioritised strategies.**   At the group level, the health centre group had variation in the priority between the HCWs group and the patients and community members group. The HCWs prioritised training while the community members prioritise dedicated infrastructure for the provision of services. The participants' rationale for selecting each strategy is italicised. (Table 2).

At the hospital group level, the commonly prioritized strategy was refresher training among the nurses-midwives, clinicians, and HSAs while the managers prioritized a change in clinic operations (Table 3).

**Consensus level at the larger group.**   At the point of consensus, the priority strategy at the health center level was refresher training for the HCWs while at the hospital level, the participants divided the strategies into short- and long-term strategies and their short-term strategy

**Table 2. Prioritised strategies at the health centre level.**

| Group | Priority 1 | Priority 2 | Priority 3 |
|---|---|---|---|
| Health care workers | Refresher training for HCWs<br>• *for the reason that with new knowledge they will provide care that is of good quality to the service users* | Male involvement.<br>• *a man is considered the head of the family and makes decisions.*<br>• *for easy management of cases for example, when HIV tests are positive, they will both be assisted*<br>• *male involvement also promotes love and bonding in families* | Clinic operations<br>• *especially task shifting to HSAs will help reduce the workload on health care workers hence providing care of good quality and at an appropriate time.* |
| Patients/ community members | Dedicated infrastructure for services<br>• *so that people are not disturbed/confused when it's time for their appointments* | Refresher training for HCWs<br>• *it helps health care workers provide care of good quality* | Clinic operations<br>• *especially on time management by healthcare workers so that patients do not spend a lot of time at the facility.* |

was working on clinic operations while refresher training was a long-term strategy (Table 4). This division was guided by the facilities' assessment of what could be easily implemented without demanding many resources.

## Discussion

In this study, we found that there are demand and supply sides as well as crosscutting strategies for improving utilisation of PNC services. Our study is different from previous studies because it employed a qualitative inquiry from multiple stakeholders including men, postnatal women, elderly, health surveillance assistants, and health professionals, and used various methods of collecting data and prioritising the strategies such as a qualitative inquiry and NGTs respectively. The demand strategies include appointment date reminders, provision of free health passport books, community awareness campaigns, and involvement of men in the services. The supply strategies included training health providers, improving clinic operations: task-

**Table 3. Prioritised strategies at the hospital level.**

| Group | Priority 1 | Priority 2 | Priority 3 |
|---|---|---|---|
| **Nurses** | Refresher Training<br>• *Because the only training health workers get is the one from their preservice* | Infrastructure for services- have a conducive environment.<br>• *Sometimes a full protocol of PNC is not provided because of limited infrastructure.* | Clinic Operation<br>• *The way the services are delivered makes it difficult for a postnatal mother to access other services*<br>*Sometimes service provision starts very late and there is no link with other services* |
| **Clinicians** | Refresher Training<br>• *The only time they trained in postnatal care is during preservice and rarely is training provided in-service* | Operational Hours to avoid long waiting times/ Appointment date reminders- feasibility to be explored.<br>• *If long waiting hours are reduced, it can motivate more women to show up*<br>*Women forget their appointment dates hence need for the reminders.* | Male Involvement<br>• *Men are key decision makers and they have been left behind, yet they can promote the use of PNC services* |
| **Community including HSAs** | Refresher Training<br>• *The only time they trained in postnatal care is during preservice* | Proximity<br>• *There a is need to shorten the distance that women have to cover.* | Community Awareness<br>• *if the community is aware of PNC, they will support the women* |
| **Matrons (Managers)** | Clinic Operations (without task-shifting and provision of free health passport books)<br>• *The way clinics are run can easily be changed so that they open early. We can't allow others beyond nurses/ midwives to conduct PNC assessments.*<br>• *The provision of free health passport books is not feasible and requires financial support from somewhere* | Training<br>• *This is important but will require a lot of preparatory work* | Proximity<br>• *This is important but will require adding more personnel and resources to run mobile clinics* |

**Table 4. Consensus strategies at health centre and hospital level.**

| | Health Centre Level | Hospital-Level |
|---|---|---|
| **Priority 1** | Refresher training for HCWs<br>• *with new knowledge, they will provide care that is of good quality to the service users* | a. Refresher training for HCWs- (this was classified as a long-term goal)<br>*Refresher Training- the right information; identification of gaps*<br>b. Clinic Operations–*limited knowledge about the clinic operations needs to be addressed first. Need to revisit the operational hours and aim at opening early.* |
| **Priority 2** | Dedicated infrastructure for services<br>• *so that people are not disturbed or confused when it's time for their appointments and privacy* | Proximity of clinics including Outreaches-<br>*Review of Community Midwives Assistant's Role and to be included in CBMNHC to be strengthened.* |
| **Priority 3** | Clinic operations<br>• *especially on task shifting to HSAs, it will help reduce the workload on health care workers hence providing care of good quality and at an appropriate time.*<br>• *especially on time management by healthcare workers so that patients do not spend more time at the facility.* | Community awareness |

shifting and hours of operation, having infrastructure for the services, and linkage to other services. Having services delivered near end-user residences was a crosscutting strategy. Additionally, we found that priority strategies include refresher training and improvement in the clinic operations especially on hours of operation, appointment date reminders, and linkage to care.

Our findings that specify a refresher training for HCWs cement previous results that have asserted that training on PNC services is a measure for improving the uptake of services [24]. Our findings build on results from an earlier review where skilled birth attendants complained of not being adequately trained to manage PNC services [8]. Furthermore, another integrative review concluded that poorly trained HCWs deter the utilisation of PNC services because their clients lose trust [25]. The provision of training could be viewed as a quest to guarantee quality PNC services which is another strategy recommended in the literature and has yielded improved rates of PNC attendance [26]. These pieces of training could be embedded as part of continuous professional development (CPD) for the HCWs. The nursing and midwifery section has a vibrant CPD platform in Malawi that can be leveraged for these pieces of training. The training could focus on the optimal points of offering postnatal care [12] as well as highlighting the PNC needs of vulnerable populations [27]. Although not mentioned in our study, a previous Malawian study stated the lack of clear guidelines as a barrier to the utilization of PNC services [28] and the proposed training could also cover such guidelines.

Task shifting of activities is common in health services [29] and could be extended to PNC services with proper training and designation of roles [11, 12]. As part of task shifting while learning from other programs like HIV and AIDS, mentors could be used in the same manner they have been used in HIV services where other HIV-infected women support pregnant and postnatal women with their initiation and adherence to ARVs [30]. In this case, then, other women could be trained to support other postpartum women with prior training and boundaries set in advance. Evidence shows that community-based health workers (CHWs) are integral to the provision of PNC services and their participation in the delivery has increased uptake [26]. Adoption of community midwifery assistants in Malawi could be a link that can be strengthened while including CHWs as well to take up some of these roles [28] with prior training on the specific postpartum assessments they need to conduct for the mother and newborn [10, 31]. Malawi has a pool of trained community midwifery assistants whose key roles

include improved access to services in the rural areas and these could serve this purpose [32]. An earlier review highlights the implementation of home visits to postpartum mothers and their newborns by trained community HCWs [10] and the use of such outreach workers has previously been reported as effective in optimising the uptake of postnatal services [33]. The use of existing CHWs needs to be taken with caution because they are usually overloaded with other concomitant responsibilities which limits the time and roles that they can play, this can be resolved by involving the community so that they strengthen the community health systems available in their setting [26].

Establishing PNC services closer to where the targetted population resides will promote utilisation because it will shorten the distance [24]. Specifically for Malawi, PNC services could be incorporated into the village health clinics and outreach clinics like under-five clinics [28]. To realize that goal, community-based HCWs should be trained for them to provide the required PNC services. As a measure to curb the distance, in Uganda, the provision of transport vouchers resulted in an increased proportion of women taking up PNC services [26]. Although vouchers seem plausible, their implementation needs to be planned with sustainability measures in place because they may prove expensive to maintain [26].

Consistent with previous findings, the lack of resources affects the utilization of PNC services [8, 28, 34]. Our study showed that women and babies attended PNC more when they received health passport books for free. This initiative could be rolled out again and studied for its impact on PNC attendance. This initiative may be Malawi-specific since Malawi uses health passport books in all its public facilities. A health passport book is a paper-based, patient-kept portable medical record used in Malawi to record the care clients including pregnant women and their babies get during their hospital appointments and it costs between 300 – 500MWK (0.29 to 0.49 USD). Initially, these were provided free to every child until the project that supported the initiative ended.

Our study recommended clinic appointment reminders for women which may be a challenge in rural settings. This area requires more research to find ways of implementing it cognizant that in Malawi only 43% of the population own mobile phones with more males (44.9 percent) compared to females (37.7 percent) owning phones [35] which may impede sending short text messages as reminders. Another strategy that can be adapted from the management of the immunization program would be the use of a town crier who can remind women about visits to PNC using a megaphone in prominent places in the community [36, 37]. Town criers have been suggested for alerting pregnant women about eclampsia [36] and general maternity issues [38]. We recommend that health facilities should leverage the mass media campaigns that occur in the communities [39] that are on maternity issues and should deliberately emphasize the relevance and time points of postnatal visits.

The involvement of male partners as a strategy for improving the uptake of PNC services as suggested in this study is consistent with earlier findings that have advocated for the inclusion of men and other birth partners [13]. This support could be tapped from the other family members who could be key in decision making as shown in some cultures [40]. Strengthening community structures and awareness is another strategy for improving the utilisation of PNC services [26, 28]. A recent review advocates for postnatal education in the form of a structured predischarge information sheet that informs a woman and her family on the postnatal care needs of the woman and the newborn, which is further strengthened post-discharge [41]. This form of education is likely to achieve uniformity in the key messages and practices that a woman and her family receive with the potential of being reinforced by the family members [10, 42].

The aspect of linking postnatal mothers and their newborns to available services is closely linked to previous calls on integrating services so that services are accessed in one spot [13].

Integration of services achieves client-centred care which helps in promoting quality care that matches the individual's needs [43]. The use of community awareness as a strategy for increasing postnatal care utilisation as highlighted in our study resonates with findings from a recent review that asserts the same with more emphasis on the rural communities and among women with low education levels to improve their knowledge and attendance to PNC [10].

### Strengths and limitations

The use of multiple methods of collecting data from various types of participants enables us to achieve a wider scope of decision-makers in postnatal care services. The prioritization exercise engaged local stakeholders to highlight the key strategies that could be feasible to implement in the area.

Our findings should be interpreted that this was a qualitative study that occurred in one district such that they may only be applicable in that setting. Although some study activities were delayed secondary to COVID-19, the researchers managed to gather all the necessary data by delaying nominal group techniques to comply with COVID-19 preventive measures.

Our study did not narrow down to a specific time point along the cascade of PNC services. Future studies should explore if the strategies would be effective at all time points. This study did not explore the depth of cultural beliefs to suggest strategies at that level, thus, future studies could focus on this in greater detail and develop strategies that are aligned with that.

### Conclusion

Optimisation of postnatal care services will require the implementation of strategies that are acceptable and relevant in the context where services are provided. The strategies for improving the uptake of postnatal care services require a multisectoral approach of both community and health system-based interventions. The health system could use a combination of short and long-term strategies to improve the uptake of services. The organisation and implementation of PNC clinic activities is a simple strategy that health facilities can implement to maximise attendance. The current education channeled towards the HCWs and the community on maternity services should include an awareness of postnatal services that are tailored to address the gaps as outlined by the intended providers and users of the services. Further research should focus on evaluating the effectiveness of the stated strategies in improving the uptake of PNC and the factors that are associated with their implementation.

### Supporting information

**S1 Text. Focus group discussion guide.**
(DOCX)

**S2 Text. In-depth interview guide.**
(DOCX)

**S3 Text. Characteristics of participants.**
(DOCX)

### Acknowledgments

This study was made possible through financial assistance from UNICEF. We are thankful to the study participants for their availability and willingness to participate in the study, the research assistants who collected the data, and the Director of Health and Social Services of Thyolo for permitting us to conduct the study in the facilities.

## Author Contributions

**Conceptualization:** Alinane Linda Nyondo-Mipando, Tiyese Jean Chimuna, Martin Dohlsten, Bongani Chikwapulo.

**Data curation:** Marumbo Chirwa, Sangwani Salimu.

**Formal analysis:** Alinane Linda Nyondo-Mipando, Marumbo Chirwa, Sangwani Salimu, Jacqueline Rose Chinkonde, Tiyese Jean Chimuna, Bongani Chikwapulo.

**Funding acquisition:** Alinane Linda Nyondo-Mipando, Martin Dohlsten, Bongani Chikwapulo, Mesfin Senbete.

**Investigation:** Alinane Linda Nyondo-Mipando, Marumbo Chirwa, Sangwani Salimu, Andrew Kumitawa, Jacqueline Rose Chinkonde, Martin Dohlsten, Bongani Chikwapulo, Tedbabe D. Hailegebriel, Debra Jackson.

**Methodology:** Alinane Linda Nyondo-Mipando, Martin Dohlsten, Debra Jackson.

**Project administration:** Jacqueline Rose Chinkonde, Tiyese Jean Chimuna.

**Resources:** Tiyese Jean Chimuna, Bongani Chikwapulo, Mesfin Senbete.

**Supervision:** Alinane Linda Nyondo-Mipando, Jacqueline Rose Chinkonde, Bongani Chikwapulo, Mesfin Senbete, Fatima Gohar, Tedbabe D. Hailegebriel, Debra Jackson.

**Writing – original draft:** Alinane Linda Nyondo-Mipando.

**Writing – review & editing:** Marumbo Chirwa, Sangwani Salimu, Andrew Kumitawa, Jacqueline Rose Chinkonde, Tiyese Jean Chimuna, Martin Dohlsten, Bongani Chikwapulo, Mesfin Senbete, Fatima Gohar, Tedbabe D. Hailegebriel, Debra Jackson.

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
