## [Decision Letter · Decision Letter 0]

4 Dec 2023

PGPH-D-23-01545

Exploring and prioritising strategies for improving uptake of postnatal care services in Thyolo, Malawi: a qualitative study.

Dear Dr. Nyondo-Mipando,

Thank you for submitting your manuscript to PLOS Global Public Health. After careful consideration, we feel that it has merit but does not fully meet PLOS Global Public Health’s publication criteria as it currently stands. Therefore, we invite you to submit a revised version of the manuscript that addresses the points raised during the review process, those you may find below under "Reviewer's Comments".

Guidelines for resubmitting your figure files and Journal Requirements are available below the reviewer comments at the end of this letter.

We look forward to receiving your revised manuscript.

Kind regards,

Rashed Shah, DrPH, M.Sc., MBBS

Academic Editor

Journal Requirements:

**Reviewers' comments:**

**Reviewer #1: **

Congratulations to the authors for the great effort in putting this paper together. Some few grammatical errors regarding punctuations noted across the work. I recommend minor revisions to the work prior to publication. Please consider the comments below for revision.

ABSTRACT:

1. Your abstract is too long, standard word limit should not exceed 250 words. Please consider editing the abstract to not more than 250 word count.

2. Insert a "full stop" at the end of the first sentence under methods in the abstract.

KEY WORDS:

Consider deleting the "coma" punctuation at the end of the last key word.

INTRODUCTION:

1. Paragraph 1, sentence 3, intext citation appears as superscript inappropriately written

2. Paragraph 2, 2nd but last sentence, normal temperature ranges between 36.5 to 37.5 and not 35.5

3. Consider inserting "and" before hyperbilirubinemia in the last sentence

4. Paragraph 4, Average days should be approximated to hours and not decimals. For example 0.5days could be written as 12hours

5. Paragraph 5, percentage of PNC visits of mothers and newborns within 48hours (84% and 88% respectively) contradicts with the less than 50% PNC attendance reported in the abstract of the manuscript. Secondly, these percentages does not depict a serious health challenge that requires investigation. Which is which?

METHODOLOGY:

A. Design: Delete punctuation mark "coma" before (HSA) in sentence 4.

B. Setting: Correct the grammatical error and insert punctuation mark appropriately in the last sentence

C. Sampling and Sample size: More information is required in this section

1. What was the total sample size for the study and how was the total sample size determined for IDIs and FGDs? Please include that in the writeup.

2. Please be consistent with the numbering system, either write in numerals or in words. For example use either 1, 2, 3 or one, two, three

3. What were the inclusion and exclusion criteria for selecting participants?

RESULTS: Additional information is required in this section.

1. How many themes and subthemes emerged from the analysis? This could be stated or tabulated for understanding.

2. Be consistent in writing participant IDs after each quotation. For example you could chose to stick to the format of (Facility, category of participant, and means of data collection).

DISCUSSION:

1. Consider inserting a "full stop" at the end of the first sentence

2. Inappropriate font use, consider editing font type and size in the 5th paragraph (3rd but last sentence)

CONCLUSION:

Consider inserting a "full stop" at the end of the first sentence

Thank you

**Reviewer #2: **

The manuscript is technically sound, and the data support the conclusions. It provides a description that is methodologically and ethically rigorous. No statistical analysis is applicable because the approach is qualitative.

**Guidelines for resubmitting figure files **

**Journal Requirements:**

---

## [Decision Letter · Decision Letter 1]

13 Feb 2024

Exploring and prioritising strategies for improving uptake of postnatal care services in Thyolo, Malawi: a qualitative study.

PGPH-D-23-01545R1

Dear Assoc. Prof Nyondo-Mipando,

We are pleased to inform you that your manuscript 'Exploring and prioritising strategies for improving uptake of postnatal care services in Thyolo, Malawi: a qualitative study.' has been provisionally accepted for publication in PLOS Global Public Health.

Best regards,

Julia Robinson

Executive Editor

Reviewer Comments (if any, and for reference):

Reviewer's Responses to Questions

**Comments to the Author**

1. If the authors have adequately addressed your comments raised in a previous round of review and you feel that this manuscript is now acceptable for publication, you may indicate that here to bypass the “Comments to the Author” section, enter your conflict of interest statement in the “Confidential to Editor” section, and submit your "Accept" recommendation.

Reviewer #1: All comments have been addressed

2. Does this manuscript meet PLOS Global Public Health’s publication criteria? Is the manuscript technically sound, and do the data support the conclusions? The manuscript must describe methodologically and ethically rigorous research with conclusions that are appropriately drawn based on the data presented.

Reviewer #1: Yes

3. Has the statistical analysis been performed appropriately and rigorously?

Reviewer #1: Yes

4. Have the authors made all data underlying the findings in their manuscript fully available (please refer to the Data Availability Statement at the start of the manuscript PDF file)?

Reviewer #1: Yes

5. Is the manuscript presented in an intelligible fashion and written in standard English?

Reviewer #1: Yes

6. Review Comments to the Author

Reviewer #1: (No Response)

7. PLOS authors have the option to publish the peer review history of their article (what does this mean?). If published, this will include your full peer review and any attached files.

**Do you want your identity to be public for this peer review?** For information about this choice, including consent withdrawal, please see our Privacy Policy.

Reviewer #1: **Yes: **Mustapha Mahama
